# Causes and Consequences of Coronavirus Spike Protein Variability

**DOI:** 10.3390/v16020177

**Published:** 2024-01-25

**Authors:** Fabian Zech, Christoph Jung, Timo Jacob, Frank Kirchhoff

**Affiliations:** 1Institute of Molecular Virology, Ulm University Medical Center, 89081 Ulm, Germany; 2Institute of Electrochemistry, Ulm University, 89081 Ulm, Germany; christoph.jung@kit.edu (C.J.); timo.jacob@uni-ulm.de (T.J.); 3Helmholtz-Institute Ulm (HIU) Electrochemical Energy Storage, 89081 Ulm, Germany; 4Karlsruhe Institute of Technology (KIT), 76021 Karlsruhe, Germany

**Keywords:** SARS-CoV-2, Sarbecoviruses, Spike, mutation, manifestation, immune evasion, zoonoses

## Abstract

Coronaviruses are a large family of enveloped RNA viruses found in numerous animal species. They are well known for their ability to cross species barriers and have been transmitted from bats or intermediate hosts to humans on several occasions. Four of the seven human coronaviruses (hCoVs) are responsible for approximately 20% of common colds (hCoV-229E, -NL63, -OC43, -HKU1). Two others (SARS-CoV-1 and MERS-CoV) cause severe and frequently lethal respiratory syndromes but have only spread to very limited extents in the human population. In contrast the most recent human hCoV, SARS-CoV-2, while exhibiting intermediate pathogenicity, has a profound impact on public health due to its enormous spread. In this review, we discuss which initial features of the SARS-CoV-2 Spike protein and subsequent adaptations to the new human host may have helped this pathogen to cause the COVID-19 pandemic. Our focus is on host forces driving changes in the Spike protein and their consequences for virus infectivity, pathogenicity, immune evasion and resistance to preventive or therapeutic agents. In addition, we briefly address the significance and perspectives of broad-spectrum therapeutics and vaccines.

## 1. Introduction

*Coronaviridae* represent a large family of diverse enveloped single-strand RNA viruses that received their name from the crown-like appearance of their Spike surface glycoproteins. Bats and rodents are considered the reservoir species of most coronaviruses [1]. However, coronaviruses are notorious for their ability to cross species barriers [2,3]. Consequently, they have been detected in many animal species and have been successfully transmitted to humans at least seven times [4]. Four of the seven human coronaviruses (229E, NL63, OC43 and HKU1) cause mild respiratory infections and are responsible for about 20% to 30% of common colds [5,6]. The remaining three cause severe respiratory diseases, with lethality rates ranging from approximately 1% (SARS-CoV-2) to around 10% (SARS-CoV-1) and 40% (MERS-CoV) [7,8]. SARS-CoV-2 was most likely transmitted to humans from bats or an intermediate host at the end of 2019 and caused the COVID-19 pandemic [6,9,10,11,12,13]. To date, SARS-CoV-2 has infected more than 800 million people. Reliable and affordable tests, as well as effective vaccines and therapeutics, have been developed in an amazingly short time. However, emerging new SARS-CoV-2 variants that at least partially escape immunity generated by previous infection or vaccination and/or show increased replication and transmissibility continue to circulate in the human population [14,15].

The overall high diversity of different members of the coronavirus family and the ongoing emergence of new SARS-CoV-2 variants through mutation and recombination give the impression that coronaviruses mutate and diversify very rapidly. However, their mutation and diversification rates are actually much lower compared to other RNA viruses, such as human immunodeficiency virus (HIV) or hepatitis C virus (HCV) [16,17]. The reason for this is that the coronavirus polymerase has a proofreading activity, which rectifies errors during replication—a trait rather rare in the realm of RNA viruses [18,19]. For SARS-CoV-2, mutation rates of 1.3 × 10^−6^ per base and infection cycle have been reported [17]. Thus, coronaviruses are at the low end of the spectrum of 10^−4^ to 10^−6^ nucleotide substitutions per replication cycle reported for RNA viruses and approximating those of DNA viruses, which range from 10^−6^ to 10^−8^ [16]. Proofreading activity also allows coronaviruses to possess one the largest genomes among RNA viruses, approximately 30 kilobases in length and encoding for about 30 proteins [20,21].

Mutation, recombination and ongoing replication are prerequisites for virus diversification. Usually, however, only changes providing a selection advantage for viral spread will be enriched and manifested in virus populations [22,23]. The forces driving fitness advantages vary and change upon zoonotic transmission. For example, it has been reported that SARS-CoV-2 initially evolved alterations increasing its replication and transmission fitness in the new human host [15,22]. After infection and/or vaccination of large parts of the human population, however, selection pressure to evade humoral immune responses increased and drove the emergence of new SARS-CoV-2 variants [15,22,24]. In some cases, initial changes allowing humoral immune evasion came at the cost of decreased infectivity but subsequent changes restored viral replication fitness [15,25]. The ongoing arms race between immune control and viral evasion drives the constant evolution of new SARS-CoV-2 variants.

As of December 2023, more than 13 billion COVID-19 vaccine doses have been administered (WHO COVID-19 dashboard) and most people have some immunity against SARS-CoV-2 through vaccination and/or previous infection. Thus, SARS-CoV-2 has lost most of its fear factor, although it continues to circulate and evolve within the human population. Thus, new questions arise, such as whether vaccines and therapeutics that are currently available or under development can protect us against newly emerging SARS-CoV-2 variants and future zoonoses of animal coronaviruses. Here, we address some of the factors driving the evolution of SARS-CoV-2 Spike proteins and their consequences for viral fitness and sensitivity to vaccines, neutralizing antibodies (nAbs) and Spike-targeting therapeutics. In addition, we discuss whether broad or even general protection against new SARS-CoV-2 variants and future zoonoses of animal coronaviruses may be feasible.

## 2. Early Features of the SARS-CoV-2 Spike Protein

The Spike proteins of different members of the coronavirus family show high sequence divergence [26]. SARS-CoV-2 is phylogenetically closely related to some bat CoVs, such as BANAL-20-52 and RaTG13 (Figure 1) [9,27,28]. The Spike protein of SARS-CoV-2 shows ~96% amino acid sequence homology to its closest bat relatives, with most variations located in the receptor binding domain (RBD) and N-terminal domain (NTD) (Figure 2). In line with this high variability, the seven human coronaviruses use several receptors in the human host [29,30]. For example, the seasonal hCoVs 229E and OC43 use aminopeptidase N (hAPN) or *N*-acetyl-9-*O*-acetylneuraminic acid, respectively, for entry into their target cells [31,32]. In comparison, the highly pathogenic MERS-CoV utilizes the dipeptidyl peptidase 4 (DPP4), an exonuclease present in cells of the upper airway and kidney, for infection [33]. Attachment and entry of MERS-CoV are supported by a cellular cofactor, CEACAM5 [34]. Recently, the transmembrane serine protease 2 (TMPRSS2) was identified as the cellular receptor of the seasonal hCoV-HKU1 [35]. Notably, TMPRSS2 is also critical for efficient SARS-CoV-2 infection, since it activates the Spike protein at the S2’ site to release the fusion peptide during the entry process [36,37,38]. Just like the Spike proteins of SARS-CoV-1 and hCoV-NL63, SARS-CoV-2 utilizes the angiotensin-converting enzyme 2 (ACE2) receptor for the infection of human target cells [36,39,40,41]. The Spike proteins of several coronaviruses detected in bats and potential intermediate hosts efficiently use human ACE2 for infection [42,43,44,45]. In comparison, the original RaTG13 Spike is poorly infectious in human cells, but a single T403R change allows efficient usage of ACE2 receptors from various species including humans [43,46]. Most Spike proteins of related bat coronaviruses contain an R or K residue at position 403 of the viral Spike protein and are capable of infecting human cells. Thus, the precursor of SARS-CoV-2 was most likely able to utilize human ACE2 before cross-species transmission. A variety of alternative receptors have been reported to promote ACE2-independent entry into human cells [47]. These include C-type lectins and phosphatidylserine receptors that promote the entry of a wide range of viruses and should thus be considered as attachment rather than entry factors [48]. Recently, it has been reported that TMEM106B, a lysosomal transmembrane protein, allows SARS-CoV-2 entry in the absence of ACE2 [49]. Currently, the relevance of attachment factors and receptors that might allow ACE2-independent infection for SARS-CoV-2 replication and transmission in vivo is poorly understood [47].

The SARS-CoV-2 Spike protein is characterized by two proteolytic cleavage sites, the S1/S2 and the S2’ site (Figure 2), which may have played a significant role in SARS-CoV-2 transmission and evolution [50,51,52]. The S1/S2 cleavage site is located at the boundary of the S1/S2 subunits of the Spike protein and distinguishes SARS-CoV-2 from related animal coronaviruses [41,53,54]. This polybasic site comprises an insertion (680SPRRAR↓SV687), forming a cleavage motif (RxxR) for furin-like enzymes, enabling proteolytic activation of the Spike protein and thus promoting entry into host cells [51]. The polybasic site allows Spike cleavage during virus packaging, thereby significantly enhancing viral transmissibility and expanding its tissue tropism [54]. The furin cleavage site enhances the ability of SARS-CoV-2 to infect certain cell types and induce cell–cell fusion, which may promote efficient viral spread [55]. Thus, it is thought that this site played a role in the rapid spread of the COVID-19 pandemic. The origin of the S1/S2 furin cleavage site in the SARS-CoV-2 Spike, and particularly the question of whether it was present before or after zoonotic spillover, has raised significant interest and is still under debate [56,57,58]. Some animal coronaviruses carrying a polybasic furin cleavage site in their Spike protein have been reported [45,52]. However, furin cleavage sites are not present in pangolin or bat coronaviruses that are closely related to SARS-CoV-2 [57]. Thus, it has been suggested that this specific cleavage site developed early in the process of the virus adapting to its human host [57], although the possibility that it predisposed an animal virus for efficient zoonotic transmission cannot be excluded [52]. Several SARS-CoV-2 variants of concern (VOCs) manifested mutations near the S1/S2 furin cleavage site that altered Spike processing efficiency. For instance, the early Alpha and all subsequent VOCs acquired the P681H mutation, increasing cleavage efficiency [59]. Additionally, the Omicron-specific N679K mutation hampered Spike processing and conferred a shift toward upper airway replication in hamster models [60].

The second step of proteolytic activation is mediated by S2’ cleavage. It liberates the fusion peptide of the S2 subunit, enabling its insertion into the cellular membrane and subsequent formation of the six-helix bundle, mediating fusion of the viral envelope with the host cell membrane, which is a crucial step for viral entry into the host cell [61,62,63]. Two main types of host cell proteases, Transmembrane Serine Protease 2 (TMPRSS2) and Cathepsins, are involved in this process, and each plays a distinct role depending on the cellular entry pathway of the virus [41,48,64,65]. When SARS-CoV-2 binds to the ACE2 receptor on the host cell surface, TMPRSS2 cleaves the Spike protein at the S2’ site [36]. In contrast, Cathepsins become involved when the virus enters cells via the endosomal pathway and fusion is triggered by the acidic environment of the endosome. Notably, it has been reported that the less efficient Spike cleavage of Omicron BA.1 Spike at S1/S2 is associated with reduced dependency on TMPRSS2 and a shift toward Cathepsin-dependent endosomal entry [66,67]. This shift in cellular tropism away from TMPRSS2-expressing cells is largely mediated by a H655Y substitution in Spike and may impact viral pathogenesis [38,66,68].

Both the S1/S2 and S2’ cleavage sites of the Spike protein play a crucial role in the ability of SARS-CoV-2 to infect human cells and have been a focal point in understanding the virus’s transmission dynamics and pathogenicity. The evolution of these sites in different VOCs may have significant implications for transmissibility, disease severity and vaccine efficacy. Thus, continued monitoring and research into these mutations are essential for managing the pandemic and developing effective countermeasures. In addition to using ACE2, cellular protease, attachment factors and alternative coreceptors, the Spike protein of SARS-CoV-2 has been shown to hijack usually antiviral IFITM2 and IFITM3 proteins for efficient infection [69,70,71,72]. Furthermore, it has been reported that the SARS-CoV-2 Spike counteracts the restriction factor tetherin that otherwise inhibits the release of viral particles [73]. Recent evidence demonstrates that mutations in Omicron Spike proteins increase their ability to counteract tetherin [74]. For the most part, however, it remains to be determined whether these mechanisms are conserved in the coronavirus family and to what extent they contributed to the efficient spread of SARS-CoV-2.

## 3. Initial Human Adaptation of SARS-CoV-2 Spike Proteins

Common cold coronaviruses, which circulate in the human population, are highly divergent from SARS-CoV-2. In contrast, SARS-CoV-1 Spike proteins share about 76% homology with that of SARS-CoV-2 and might induce cross-neutralizing antibodies [75]. However, SARS-CoV-1 infected only ~8000 individuals and has fortunately disappeared. Thus, SARS-CoV-2 essentially hit an immunologically naïve population, and initial adaptations to humans manifested in changes that increased viral infectivity and transmission. For example, the mutation of D614G in the C-terminal region of the S1 domain of the Spike became prevalent during the first few months of the pandemic, indicating a significant selective advantage for the spread of SARS-CoV-2 in human populations [76,77,78,79]. It has been reported that D614G alters Spike configuration and enhances viral replication in human cells, as well as in the human respiratory tract, hence increasing transmission rates but not pathogenicity [78,79,80,81,82]. Another mutation that emerged relatively rapidly is N501Y in the RBD, which increases binding of the Spike protein to the ACE2 receptor [83,84]. Additionally, evolution of SARS-CoV-2 codon usage and the slower-than-expected acquisition of mutations, hinting at a purifying selection during the initial phase of the pandemic, provide insight into the virus’s genomic adaptation strategies [85,86]. Alterations in the RBD of the Spike protein, which increased the affinity for the human ACE2 receptor, proteolytic processing and fusion were suggested to represent adaptive steps critical for efficient human-to-human transmission of SARS-CoV-2 [48,87,88,89]. Since its emergence, thousands of mutations have been observed, with novel variations continuously emerging as the virus replicates and spreads across the human population [90].

## 4. Evasion of Adaptive Immunity

After SARS-CoV-2 infection and/or vaccination of significant parts of the human population, selection pressures shaping the evolution of the Spike shifted from alterations increasing viral infection and replication fitness to mutations allowing evasion of humoral immune responses [15,22]. The initial variants of concern (VOCs), named Alpha (B.1.1.7), Beta (B.1.351), Gamma (P.1) and Delta (B.1.617.2), all emerged independently [15]. Each of these variants contained about six to eight changes in the Spike protein, most of them in the RBD (Figure 2), promoting mainly immune evasion. However, changes affecting ACE2 binding and increasing fusogenicity have also been reported [15,91]. This has changed with the emergence of the Omicron VOCs, which contained a strikingly high number of changes, especially in the viral Spike protein, compared to all previous SARS-CoV-2 variants [92]. The initial Omicron BA.1 VOC outcompeted the previously dominating Delta VOC at enormous speed, although it displayed low infectivity and replication fitness in many cell culture systems and animal models [38,67,93,94,95]. It was itself outcompeted by BA.2, which differs by ~20 amino acid changes in Spike from BA.1. All subsequent and current VOCs originated from BA.2 and contain further amino acid changes in their Spike proteins. Accumulating evidence suggests that initial mutations facilitated viral immune evasion at the cost of reduced infectivity and subsequent changes restored infectiousness and replication fitness [25,96]. New variants of Omicron are constantly emerging showing mutations in the RBD, representing the main target of neutralizing antibodies and the N-terminal domain (NTD) of Spike and allowing immune evasion [97,98,99,100] (Figure 2), thus leading to the simultaneous emergence of sub-variants; each variant is characterized by mutations that converge on several hotspots of their RBDs. Specific strains like BQ.1.1.10, BA.4.6.3, XBB and CH.1.1 were identified as highly antibody-evasive [99,101]. This phenomenon of convergent evolution is driven in part by the humoral immune pressure, which promotes the evolution of the virus in a way that helps it evade nAbs [102]. Just like common cold coronaviruses, infection with one strain of SARS-CoV-2 does not efficiently protect against infection with another strain or newly emerging variants. For example, the XBB lineages that currently dominate the pandemic are largely resistant against neutralization via humoral immune responses induced by infection with earlier SARS-CoV-2 variants (including BA.1, BA.2 and BA4/5) or vaccination [103,104,105,106]. The SARS-CoV-2 variant of interest, BA.2.86, that was first isolated in July 2023, has 36 amino acid substitutions compared to XBB.1.5, many of them located in key antigenic sites of the Spike protein. The proportion of BA.2.86 and closely related descendent lineages, such as JN.1, characterized by an additional substitution of L455S in the Spike protein, is currently steadily increasing, indicating high transmission fitness and efficient humoral immune evasion [107]. However, T cell responses may remain effective and prevent severe disease in most cases [108,109]. Altogether, SARS-CoV-2 continues to infect humans but in relatively stable numbers, and has transitioned from the pandemic to the endemic phase [110,111].

## 5. Broadly Acting Vaccines or Therapeutics Targeting the SARS-CoV-2 Spike Protein

The evolution of new SARS-CoV-2 variants also conferred resistance to therapeutic antibodies and vaccines. For example, neutralizing antibodies (nAbs) against SARS-CoV-2 initially inhibited a variety of virus strains and showed great promise in treating and preventing infections [112]. The initially developed nAbs target epitopes in the RBD overlap with the ACE2 receptor-binding site (RBS), thus sterically hindering binding of the Spike glycoprotein to its receptor. However, emerging SARS-CoV-2 variants show resistance to essentially all first generation FDA-approved monoclonal antibodies (Figure 3) [113,114,115]. Specifically, mutations of E484A/K and Q493R in the RBD of the Spike protein rendered SARS-CoV-2 resistant to Bamlanivimab and N440K, and G446S to Imdevimab [96,116]. The Omicron variants have shown an impressive capability to evade even those nAbs that target more conserved domains in the Spike RBD region [117,118]. However, while coronavirus Spike proteins are highly variable and tolerate numerous changes in their RBD, some structural and functional features are highly conserved, offering perspectives for broad-spectrum antiviral agents.

Broadly neutralizing antibodies (bnAbs) designed to target highly conserved regions of the S2 region of the Spike protein, including the fusion peptide and Stem-helix regions, combine the potential for pan-coronavirus activity with a high genetic barrier to evasion [119,120]. S2 stem helix-binding bnAbs, isolated from SARS-CoV-2 recovered-vaccinated donors, showed broad cross-protection against SARS-CoV-1, SARS-CoV-2 and MERS-CoV in mouse models [119]. The bnAbs targeting the SARS-CoV-2 Spike fusion peptide not only reduced viral fusion, but additionally impaired proteolytic maturation of the Spike glycoprotein [121]. Thus, bnAbs targeting conserved domains in the S2 region of the viral Spike protein are promising candidates for next-generation pan-coronavirus vaccine development.

Peptide inhibitors are increasingly recognized in antiviral drug development due to their high specificity and biocompatibility [122]. Similar to pan-coronavirus bnAbs, targeting conserved domains and fusion mechanisms of the SARS-CoV-2 Spike S2 domain allows the development of pan-coronavirus antiviral peptides. Peptides or small proteins derived from the helix H1 region of ACE2 mimic key elements of ACE2 interaction with the viral Spike protein [123,124,125,126]. These compounds competitively inhibit Spike binding to the ACE2 receptor, thereby preventing viral entry [123]. The utilization of ACE2 as a primary receptor is conserved between all SARS-CoV-2 VOCs. Although alternative receptors for SARS-CoV-2 entry have been reported, development of resistance to ACE2 mimics by a switch in receptor usage seems unlikely. Thus, agents mimicking ACE2 offer a promising avenue for the development of effective anti-SARS-CoV-2 treatments.

Just like many other enveloped viruses, such as HIV-1, entry of SARS-CoV-2 requires the insertion of a fusion peptide into the cellular membrane and the subsequent formation of a six-helix bundle between heptad repeat regions (HR1 and HR2) that pull the viral and cellular membranes together to mediate fusion. For instance, the peptide EK1 interacts with the highly conserved HR2 domain of the S2 subunit of the Spike protein (Figure 2), thereby preventing the interaction between HR1 and HR2 and, therefore, virus–cell fusion [127]. Interestingly, EK1 and its optimized derivative (EK1C4) not only display broad or even pan-activity against coronaviruses but even show cross-activity against HIV [128]. The mutation of N969K in Omicron Spike proteins induces substantial changes in the structure of the HR2 backbone in the HR1/HR2 post-fusion bundle. Nonetheless, EK1 and EK1C4 inhibit membrane fusion, mediated by the Spike proteins of Omicron subvariants [129]. In addition, inhibitors of the proteases activating coronavirus Spike proteins, i.e., furin, TMPRSS2 and Cathepsins, may offer perspectives for broad-based inhibitors [130,131,132].

The initial vaccines developed against SARS-CoV-2 were designed based on the Hu-1 variant, the earliest strain of the virus [133,134,135]. These vaccines, including mRNA-based vaccines like BNT162b2 (Pfizer, New York, NY, USA—BioNTech, Mainz, Germany) or mRNA-1273 (Moderna, Cambridge, MA, USA), vector-based vaccines like Vaxzevria (AstraZeneca, London, UK) or inactivated virus vaccines like CoronaVac (Sinovac, Beijing, China) were highly effective in reducing COVID-19 pathogenicity and mortality [136,137]. In response to the highly immune, evasive SARS-CoV-2 Omicron BA.4/BA.5 and XBB variants, adapted vaccines, like the bivalent Spikevax (Moderna) or the Omicron XBB1.5-adapted variants of BNT162b2, were designed to better match circulating strains [138]. Authorized for use, these vaccines aim to offer broader protection against COVID-19, preventing hospitalization and death due to infection. First studies demonstrate the enhanced effectiveness of these bivalent boosters against currently circulating SARS-CoV-2 variants compared to the original monovalent vaccinations [138]. Several strategies have been pursued to induce broad protection against diverse coronaviruses. For example, a trivalent sortase-conjugate nanoparticle vaccine that contained RBDs from SARS-CoV-2, RsSHC014 (a bat coronavirus) and MERS-CoV elicited neutralizing antibody responses against these viruses [139]. Another approach is based on mRNA vaccines designed to express chimeric Spike proteins, aiming to elicit protection against a range of *Sarbecoviruses*, including SARS-CoV, SARS-CoV-2 and various bat coronaviruses [140]. However, neither previous coronavirus infections nor current vaccines confer long-term protection against all newly emerging SARS-CoV-2 variants. Thus, whether broad and long-lasting protection by alternative vaccination strategies can be achieved remains to be determined.

## 6. Conclusions and Future Perspectives

Coronaviruses have already been successfully transmitted from animals to humans on at least seven independent occasions. Furthermore, exposure to diverse *Sarbecoviruses* has been identified among high-risk human communities, providing epidemiological and immunological evidence that zoonotic spillover is continuously occurring [141]. Thus, broadly effective and long-lasting therapeutic and preventive agents are highly desirable. The rapid adaptation and evolution of coronaviruses necessitate either a proactive approach in developing therapeutics and vaccines or the targeting of highly conserved domains and/or mechanisms. Human immune responses and newly emerging SARS-CoV-2 variants are in a constant arms race. Notably, even the most recent and divergent SARS-CoV-2 variants show only 3.3% (43 mutations in Omicron XBB1.5) to 3.5% (45 mutations in Omicron EG.5.1) amino acid diversity from the Spike proteins of the early virus strains. This is sufficient to confer efficient resistance against humoral immune responses induced by previous infection or vaccination. In comparison, bat coronaviruses and SARS-CoV-1 showing up to 30% amino acid diversity in Spike from SARS-CoV-2 are efficiently neutralized [43,75,142,143]. However, most mutations in SARS-CoV-2 VOCs, especially Omicron, are located in the RBD region of the viral Spike protein that is the main target of neutralizing antibodies. This illustrates the enormous power of selection pressures in humans, driving changes in the viral Spike proteins that enable efficient immune escape and highlight the importance of therapeutically targeting the highly conserved regions of the Spike protein. Much has been learned from the intense research effort on SARS-CoV-2 and other viral pathogens with regard to the development of broad-spectrum or pan vaccines and therapeutics against coronaviruses. As outlined above, epitope-optimized vaccines inducing antibodies directed against conserved regions of the Spike protein, ACE2-derived mimetics, peptides targeting the HR1 domain in the S2 subunit of Spike to prevent fusion, as well as monoclonal antibodies targeting conserved domains in Spike hold great promise for broad protection against coronaviruses. This knowledge will also allow for the improvement of preventive and therapeutic strategies against other viral pathogens and hopefully help to prevent future pandemics.

## Figures and Tables

**Figure 1 viruses-16-00177-f001:**
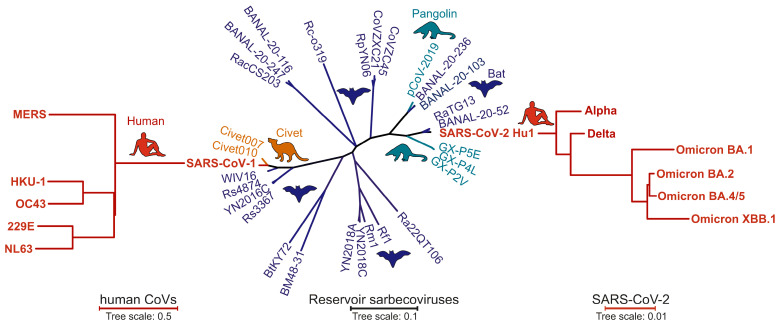
Phylogenetic relationship between human coronavirus (**left**), animal Sarbecovirus (**middle**) and human SARS-CoV-2 VOC Spike (**right**) amino acid sequences from the indicated viral strains or species. Amino acid-based sequence alignments were performed in Clustal Omega https://www.ebi.ac.uk/Tools/msa/clustalo/ (accessed on 12 December 2023) using the ClustalW algorithm and an ordered input. The resulting phylogenetic trees were transferred to ITOL https://itol.embl.de/ (accessed on 12 December 2023) and visualized as rectangular (human coronaviruses) or unrooted (Sarbecoviruses) phylogenetic trees in default settings. Host species are indicated by silhouettes: bat (blue), pangolin (turquoise), civet (orange) and human (red). Sequence identifiers are provided in Appendix A.

**Figure 2 viruses-16-00177-f002:**
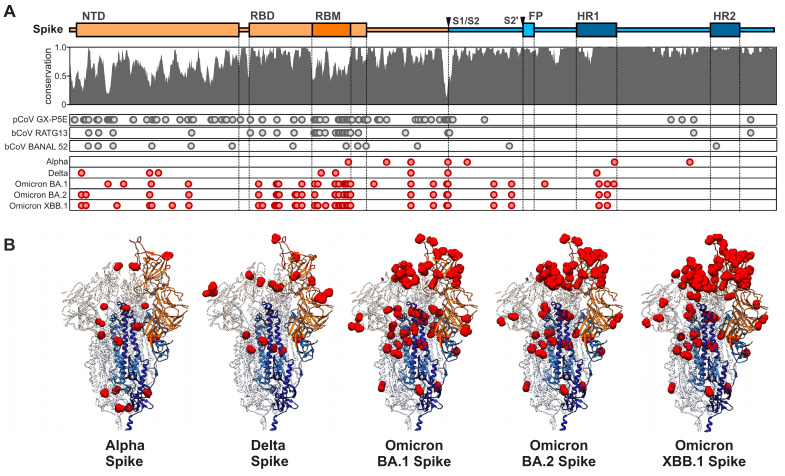
Conservation and mutation mapping of SARS-CoV-2 VOCs and Sarbecovirus Spike proteins. (**A**) Conservation of 29 Sarbecovirus Spike proteins, with regions indicated as NTD (N-terminal domain, orange), RBD (receptor-binding domain, orange), RBM (receptor-binding motif, dark orange), S1/S2 and S2’ (protease cleavage sites, black arrows), FP (fusion peptide, blue) and HR1/2 (heptad repeat 1/2, dark blue). Positions of mutations of selected bat Sarbecoviruses (gray) and SARS-CoV-2 VOCs (red) compared to the reference (GenBank: BCN86353.1) are indicated as circles. (**B**) Overview of the SARS-CoV-2 Spike structure (PDB: 7KNB) and localization of amino acid changes in the indicated SARS-CoV-2 VOCs. Color coding according to domains as indicated.

**Figure 3 viruses-16-00177-f003:**
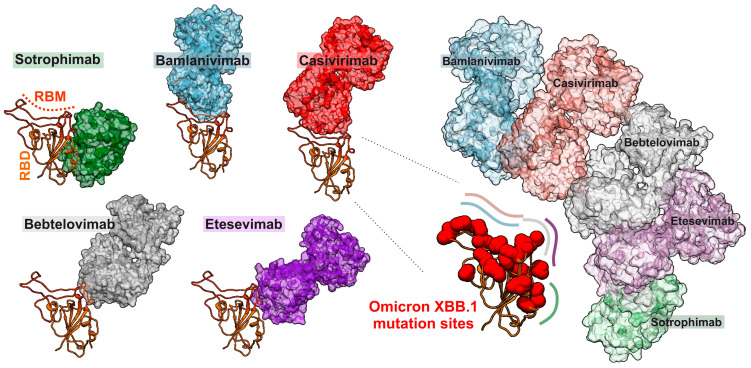
Illustration of binding sites of nAbs to the Spike RBD. The RBD of the SARS-CoV-2 Spike protein (PDB: 7KNB, yellow ribbon) and interacting therapeutic monoclonal antibodies are indicated: Bamlanivimab (blue, PDB: 7KMG), Casirivimab (red, PDB: 7M42), Bebtelovimab (gray, PDB: 7MMO), Etesevimab (purple, PDB: 7F7E) and Sotrovimab (green, PDB: 7TLY). The positions of RBD-specific mutations in SARS-CoV-2 Omicron XBB.1 are highlighted by red spheres.

## Data Availability

The raw data supporting the conclusions of this article will be made available by the authors on request.

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
