# Peer review of "Causes and Consequences of Coronavirus Spike Protein Variability"

_viruses, 2024, doi:10.3390/v16020177_

Round 1

Reviewer 1 Report

Comments and Suggestions for Authors

The manuscript includes a brief revision on the Coronavirus (CoV) spike (S) and its variability.  The authors revised some of the S structural features, showed the mutations of the major and most recent SARS-CoV-2 variants and their consequences on therapeutic antibody (Ab) recognition, and described other therapeutic proteins and vaccines.  I found that the review was clear and concise and presented the most interesting aspects related to the S variability, but it requires of substantial corrections and revisions. The conclusion section should be clarified and rewritten, perhaps highlighting the vaccines and therapeutics needed to prevent CoV transmission and overcome viral evolution.

Specific Comments:

1.   In the abstract the authors stated that the SARS-CoV-2 is the most recent zoonotic hCoV.  Nonetheless, there are no conclusive data proving that SARS-CoV-2 jumped from an animal host to humans, and it should be removed from the abstract.  As mentioned in the introduction, transmission of SARS-CoV-2 from bats to humans likely occurred through an intermediated host, but this has not been conclusively demonstrated.

2.   The statement on line 88 “ACE usage seems helpful but not essential for zoonotic transmission of coronavirus” such as SARS-CoV-1, hCoV-NL63 and SARS-CoV-2 is not accurate.  In CoV, all the described zoonotic transmissions occur through the recognition of orthologous receptors between different species and the adaptation to the human receptor in the intermediate host, as demonstrated by SARS-CoV-1.  Thus, conservation of receptor use is a key feature of interspecies transmission of CoV.  In addition, the transmission of SARS-CoV-2 to humans and its efficient human-to-human transmission indicates that the virus was able to use ACE2 prior to the zoonotic event, contrary to the statement in lines 94 and 95, which reads “immediately after”.  This statement should be deleted.  The relevance of alternative entry receptors to ACE2 has not been demonstrated and likely, they are molecules that facilitate virus entry into cells but are not necessary for cell infection.

3.   It is known that the SARS-CoV-2 XBB variant evolved from BA.2, but this is not shown in Figure 1.

4.   Figure 2: Authors should indicate the reference strain or virus used to define the mutations included in the figure.

5.   Line 170. D614G is not in the RBD.

6.   I would recommend changing Figure 3, as it is not clear how the mutations in the XBB affect individual Ab binding.  The panel on the right should show the XBB-RBD alone with all the mutations marked as red spheres, without Ab binding.  Perhaps the mutations should be indicated.  The rest of the images should also show the RBD with the XBB mutations.

7.   In the last subsection 5, the authors missed the description of small proteins designed based on the N-terminal portion of ACE2 that have very high affinity for the RBD and therapeutic potential.  See DOI: 10.1126/science.abd9909.

8.   In the conclusions, lines 309 and 310, the authors mentioned that the low mutation% in the whole spike is sufficient to confer resistance to the humoral immune response.  Nonetheless, most neutralizing Ab target the RBD, the major antigenic site in the S; the RBD among variants has a much higher mutation%, particularly in the Omicron variants that scape neutralization.  Thus, these sentences should be revised as it is more important to discuss the quality of the immune response to the RBD rather than to the S.  In addition, the sentence in lines 314-316 (“On the positive side...”) is not clear and the sentence in lines 319-320 (“This raises…”) is not precise: What animal CoV? Why is not expected to be effective?  Overall, the conclusion section is very weak, with uncertain statements, so it should be extensively revised.

9.   The authors should consider that the knowledge acquired during the pandemic is an important asset for the development of vaccines and Abs to combat new CoV outbreaks.  In addition, the conclusion should highlight the identification of Ab and other therapeutics with a broad neutralization range as well as vaccines that prevent CoV transmission among humans.

Author Response

Reviewer 1: The manuscript includes a brief revision on the Coronavirus (CoV) spike (S) and its variability.  The authors revised some of the S structural features, showed the mutations of the major and most recent SARS-CoV-2 variants and their consequences on therapeutic antibody (Ab) recognition, and described other therapeutic proteins and vaccines.  I found that the review was clear and concise and presented the most interesting aspects related to the S variability, but it requires of substantial corrections and revisions. The conclusion section should be clarified and rewritten, perhaps highlighting the vaccines and therapeutics needed to prevent CoV transmission and overcome viral evolution.

 We thank the reviewer for the positive comments and substantially revised the conclusions section to address the main concern.

Specific Comments:

  1. In the abstract the authors stated that the SARS-CoV-2 is the most recent zoonotic hCoV. Nonetheless, there are no conclusive data proving that SARS-CoV-2 jumped from an animal host to humans, and it should be removed from the abstract.  As mentioned in the introduction, transmission of SARS-CoV-2 from bats to humans likely occurred through an intermediated host, but this has not been conclusively demonstrated.

The likelihood that SARS-CoV-2 originated from a zoonosis is very high. We agree, however, that there is no direct evidence for this and replaced “zoonotic” by “human” in the abstract.

  1. The statement on line 88 “ACE usage seems helpful but not essential for zoonotic transmission of coronavirus” such as SARS-CoV-1, hCoV-NL63 and SARS-CoV-2 is not accurate. In CoV, all the described zoonotic transmissions occur through the recognition of orthologous receptors between different species and the adaptation to the human receptor in the intermediate host, as demonstrated by SARS-CoV-1.  Thus, conservation of receptor use is a key feature of interspecies transmission of CoV. In addition, the transmission of SARS-CoV-2 to humans and its efficient human-to-human transmission indicates that the virus was able to use ACE2 prior to the zoonotic event, contrary to the statement in lines 94 and 95, which reads “immediately after”.  This statement should be deleted. The relevance of alternative entry receptors to ACE2 has not been demonstrated and likely, they are molecules that facilitate virus entry into cells but are not necessary for cell infection.

We agree that the precursor of SARS-CoV-2 was most likely able to use human ACE2 and modified the text accordingly (lines 95-77). In addition, we clarified that most alternative entry factors mediate attachment rather than efficient entry (line 97-103).

  1. It is known that the SARS-CoV-2 XBB variant evolved from BA.2, but this is not shown in Figure 1

We are aware of this and mention it in the text (line 185). BA.2, BA.5 and XBB are all about equally divers from BA.1 and the branching order is not significant. We expanded the legend to Fig. 1 for clarity.

  1. Figure 2: Authors should indicate the reference strain or virus used to define the mutations included in the figure.

Good point – we added this information in the figure legend.

  1. Line 170. D614G is not in the RBD.

Corrected (lines 158-159).

  1. I would recommend changing Figure 3, as it is not clear how the mutations in the XBB affect individual Ab binding. The panel on the right should show the XBB-RBD alone with all the mutations marked as red spheres, without Ab binding.  Perhaps the mutations should be indicated.  The rest of the images should also show the RBD with the XBB mutations.

As suggested, we modified Figure 3. In the left panel, we show the RBD without mutations to illustrate how the various mAbs bind to it. In the right panel, we show the mutations in XBB without Abs. We did not label the individual mutations for clarity.

  1. In the last subsection 5, the authors missed the description of small proteins designed based on the N-terminal portion of ACE2 that have very high affinity for the RBD and therapeutic potential. See DOI: 10.1126/science.abd9909.

We added a chapter describing these interesting agents

  1. In the conclusions, lines 309 and 310, the authors mentioned that the low mutation% in the whole spike is sufficient to confer resistance to the humoral immune response. Nonetheless, most neutralizing Ab target the RBD, the major antigenic site in the S; the RBD among variants has a much higher mutation%, particularly in the Omicron variants that scape neutralization.  Thus, these sentences should be revised as it is more important to discuss the quality of the immune response to the RBD rather than to the S.  In addition, the sentence in lines 314-316 (“On the positive side...”) is not clear and the sentence in lines 319-320 (“This raises…”) is not precise: What animal CoV? Why is not expected to be effective?  Overall, the conclusion section is very weak, with uncertain statements, so it should be extensively revised.

We substantially revised the conclusions part to address these points

  1. The authors should consider that the knowledge acquired during the pandemic is an important asset for the development of vaccines and Abs to combat new CoV outbreaks. In addition, the conclusion should highlight the identification of Ab and other therapeutics with a broad neutralization range as well as vaccines that prevent CoV transmission among humans.

We agree and modified the conclusion section accordingly.

Reviewer 2 Report

Comments and Suggestions for Authors

The authors present a mini review of the causes and consequences of the SARS-CoV-2 spike protein variability. Overall, this is an easily digestible review of the topic. I have a few suggestions that may help improve the manuscript:

1. While the authors do a good job putting the SARS-CoV-2 spike in the context of other related human and animal CoV spike proteins, I think they missed an opportunity to talk about the other receptors used by human CoVs in addition to ACE2. This will help give context to the diversity of CoV entry mechanisms.

2. Line 145: There is a grammatical error in this sentence.

3. Line 173: The authors state that N501Y is "thought" to increase binding affinity to ACE2. Either the data shows that it does increase binding affinity or not. This sentence should be rephrased.

4. There is a curious lack of citations to some of the long-standing coronavirus spike protein researchers such as Tom Gallagher and Stanley Perlman. Nor is there any citation of David Veesler's group for their work on spike protein structure. This should be rectified.

Comments on the Quality of English Language

Appropriate.

Author Response

Reviewer 2: The authors present a mini review of the causes and consequences of the SARS-CoV-2 spike protein variability. Overall, this is an easily digestible review of the topic. I have a few suggestions that may help improve the manuscript:

  1. While the authors do a good job putting the SARS-CoV-2 spike in the context of other related human and animal CoV spike proteins, I think they missed an opportunity to talk about the other receptors used by human CoVs in addition to ACE2. This will help give context to the diversity of CoV entry mechanisms.

To address this, we now briefly describe which other receptors are used by other human coronaviruses (line 81-89) and mention alternative receptors of SARS-CoV-2 (line 96-103).

  1. Line 145: There is a grammatical error in this sentence.

Corrected.

  1. Line 173: The authors state that N501Y is "thought" to increase binding affinity to ACE2. Either the data shows that it does increase binding affinity or not. This sentence should be rephrased.

We agree and modified the sentence (line 163/164).

  1. There is a curious lack of citations to some of the long-standing coronavirus spike protein researchers such as Tom Gallagher and Stanley Perlman. Nor is there any citation of David Veesler's group for their work on spike protein structure. This should be rectified.

Our apologies for this. Given the huge number of publications on this topic citations are unfortunately incomplete. We fully agree that these researchers made important contributions, expanded the references, and now cite several of their key studies (new refs.13, 38, 41, 44, 45, 89, 95, 99, 100, 114, 115, 120, 126, 141)

Round 2

Reviewer 1 Report

Comments and Suggestions for Authors

The authors addressed the questions and changed the conclusion section.  I noticed that the reference sequence included in the legend of Figure 2 is not from the PDB but from the GenBank.  It is not clear why a spike sequence from Nov-2020 was used as a reference instead of a sequence from early 2020.

Author Response

We thank the Reviewer for his positive feedback. The labelling of the identifier was corrected. The
reference sequence (GenBank: BCN86353.1) was actually collected in March 2020 but uploaded to
GenBank in November 2020, as stated on the GenBank webpage.